# Genome Characterization and Probiotic Potential of *Corynebacterium amycolatum* Human Vaginal Isolates

**DOI:** 10.3390/microorganisms10020249

**Published:** 2022-01-23

**Authors:** Irina V. Gladysheva, Sergey V. Cherkasov, Yuriy A. Khlopko, Andrey O. Plotnikov

**Affiliations:** Institute of Cellular and Intracellular Symbiosis, Ural Branch of the Russian Academy of Sciences, 460000 Orenburg, Russia; cherkasovsv@yandex.ru (S.V.C.); 140374@mail.ru (Y.A.K.); protoz@mail.ru (A.O.P.)

**Keywords:** *Corynebacterium amycolatum*, vaginal microbiome, genome, secondary metabolism, bacteriocin

## Abstract

The vaginal microbiome of healthy women contains nondiphtheria corynebacteria. The role and functions of nondiphtheria corynebacteria in the vaginal biotope are still under study. We sequenced and analysed the genomes of three vaginal *C. amycolatum* strains isolated from healthy women. Previous studies have shown that these strains produced metabolites that significantly increased the antagonistic activity of peroxide-producing lactic acid bacteria against pathogenic and opportunistic microorganisms and had strong antimicrobial activity against opportunistic pathogens. Analysis of the *C. amycolatum* genomes revealed the genes responsible for adaptation and survival in the vaginal environment, including acid and oxidative stress resistance genes. The genes responsible for the production of H_2_O_2_ and the synthesis of secondary metabolites, essential amino acids and vitamins were identified. A cluster of genes encoding the synthesis of bacteriocin was revealed in one of the annotated genomes. The obtained results allow us to consider the studied strains as potential probiotics that are capable of preventing the growth of pathogenic microorganisms and supporting colonisation resistance in the vaginal biotope.

## 1. Introduction

The vaginal microbiome is an open complex multicomponent system in dynamic equilibrium [1]. The vaginal microbiome is represented by various microbial communities containing bacteria that can synthesize organic acids, including lactic acid, maintaining a vaginal pH of 3.8–4.4, thereby supporting women’s health [2,3]. It is generally accepted that the main microorganism responsible for maintaining the stability of the vaginal microbiome is the dominant microbe lactobacilli. Lactobacilli produce lactate, hydrogen peroxide and various bacteriocins and bacteriocin-like substances, thereby inhibiting the growth of obligate anaerobes and opportunistic microorganisms [2,3,4].

However, using culture-independent methods based on sequencing of the 16S (rRNA) gene, researchers demonstrated that a significant proportion (7–33%) of healthy women lack lactobacilli in their vagina [5,6]. It is known that the absence of lactobacilli is accompanied by the presence of other microorganisms, such as *Gardnerella vaginalis*, or various species of *Peptostreptococcus* spp., *Prevotella* spp., *Pseudomonas* spp., *Streptococcus* spp. and/or *Corynebacterium* spp. Such changes in the structure of the vaginal microbiome are not considered a pathological disorder [7,8].

The *Corynebacterium* genus contains approximately 130 different species of diverse and ecologically significant microorganisms. The well-known typical representatives of this genus are the pathogenic species *C. diphtheriae*, *C. ulcerans*, and *C. pseudotuberculosis* [9], and the roles of these species in the development of human infection have been proven. In addition, a large group of nondiphtheria corynebacteria is part of the resident microflora of human skin and mucous membranes and most often stands out from clinical samples [10]. For the last seven years, the number of publications on the important role of individual strains of nondiphtheria corynebacteria in protecting human mucous membranes from infection have increased. It was shown that certain types of nondiphtheria corynebacteria produce various bacteriocins, bacteriocin-like substances and biosurfactants, which inhibit the growth of opportunistic microorganisms and their biofilm formation [11,12]. Individual strains have pronounced bactericidal activity against opportunistic microorganisms, including MRSA [13]. Certain strains of nondiphtheria corynebacteria were recommended for use as probiotic microorganisms [14]. There are few reports on the use of nondiphtheria corynebacteria as immunomodulators in tumour immunotherapy [15].

Nondiphtheriae corynebacteria in vaginal biotopes were found in women regardless of age and microecological status [16,17,18]. Nondiphtheriae corynebacteria along with *Staphylococcus epidermidis* constitute the main part, approximately 80% of the vaginal microbiota, in prepubescent girls [19]. The number of nondiphtheriae corynebacteria in pregnant and postpartum women is also increased [20]. Despite the high frequency of nondiphtheriae corynebacteria occurrence in the female genital tract, studies on this topic are limited mainly to the description of pathogens [21,22].

*Corynebacterium amycolatum* was isolated for the first time by Collins and Burton from clinical specimens in 1988 [23]. Based on our observations, *C. amycolatum* is rather frequently isolated from vaginal biotopes of healthy women, and features a high probiotic potential. Particularly, we isolated three strains of corynebacteria from the vaginal contents of healthy women. All of them were identified as *C. amycolatum*. Metabolites of these strains greatly increased the antagonistic activity of peroxide-producing lactobacilli against pathogenic and opportunistic microorganisms and had strong antimicrobial activity against opportunistic pathogens such as *Escherichia coli*, *Staphylococcus aureus*, *Klebsiella pneumoniae* and *Pseudomonas aeruginosa* [24,25]. These strains showed the greatest adhesive ability to vaginal epithelial cells and human fibronectin under low pH conditions [25]. Due to their useful properties, we selected these strains for high-throughput sequencing (HTS) genome annotation. To better understand the ability of corynebacteria to survive in the vaginal biotope under eubiosis and to show their beneficial properties, we sequenced and analysed the genomes of three isolated strains. In addition, we should note that the genomic characteristics of *Corynebacterium amycolatum* have not yet been previously described.

## 2. Materials and Methods

### 2.1. DNA Preparation, Genome Sequencing and Assembly

Strains of *C. amycolatum* ICIS 5, ICIS 9 and ICIS 53 were previously isolated from vaginal smears of healthy women of reproductive age. The strains are deposited in the Collection of Microorganisms of the Institute for Cellular and Intracellular Symbiosis UrB RAS (Orenburg, Russia) under the same accession names. The phenotypic characteristics of these isolates have been previously described in detail [24,25]. The strains were kept at −80 °C in 20% (*v*/*v*) glycerol before experiment. The isolates were grown in tryptic soy broth (TSB) at 37 °C for 24 h.

Overnight bacterial cultures were used for extraction genomic DNA with the phenol-chloroform method. DNA libraries were prepared and sequenced at the Center of Shared Scientific Equipment “Persistence of microorganisms” at the Institute for Cellular and Intracellular Symbiosis UrB RAS (Orenburg, Russia). The Nextera XT DNA library preparation kit (Illumina, San Diego, CA, USA) was used according to the manufacturer’s instructions. High-throughput sequencing of the DNA libraries was carried out in the MiSeq sequencer (Illumina, USA) using the MiSeq reagent kit v3 2 × 300 cycles (Illumina, USA). The reads were quality-trimmed with the Trimmomatic tool [26]. De novo genome assembly was carried out with SPAdes (version 3.9.0) [27].

### 2.2. Genome Annotation

Functional annotation of the genomes was carried out by the RAST server (Rapid Annotation using Subsystem Technology) [28] and NCBI Prokaryotic Genome Annotation Pipeline (PGAAP) [29]. Clusters of orthologous groups (COGs) of proteins were used for functional classifications performed with the eggNOG (version 4.5) database (http://eggnog.embl.de/version_3.0/, accessed on 19 December 2021) [30]. The bioinformatic tools BAGEL4 [31] and AntiSMASH 5 [32] were used to determine potential clusters of secondary metabolites with antimicrobial activity. Antibiotic resistance genes in the genomes were predicted using the RGI (Resistance Gene Identifier) tool [33]. The presence of putative virulence genes in the genomes was investigated using the Virulence Factor of Bacterial Pathogens Database (VFDB) [34]. The CRISPR regions were identified with a CRISPR online detection tool, CRISPR finder [35].

### 2.3. Phylogenetic Analysis

Phylogenetic analysis was conducted based on the 16S rDNA sequences retrieved from draft genomes of the *C. amycolatum* strains ICIS 5 (WGS Project: SSOR01), ICIS 9 (MTPT01) and ICIS 53 (MIFV01), draft genomes of nine *C. amycolatum* strains currently available in the NCBI database (Appendix A), including SK46 (WGS Project: ABZU01), NCTC7243 (UFXE01), UMB0042 (PKHS01), UMB0338 (PKHT01), UMB7760 (VYVQ01), UMB9184 (VYVF01), UMB1182 (VYWH01), UMB1310 (VYWB01) and UMB9256 (VYVD01), and 16S rDNA sequences of the 28 type strains of *Corynebacterium* spp. most often isolated from human clinical samples and 2 strains of *Euzebya* spp. as an out group. The type strains of *Corynebacterium* spp. were selected from the List of Prokaryotic Names with Standing in Nomenclature website (http://www.bacterio.net/index.html, accessed on 19 December 2021) [36]. The 16S gene sequences of type strains were downloaded from the National Center for Biotechnology Information at https://www.ncbi.nlm.nih.gov/, accessed on 19 December 2021). Sequences were aligned by MUSCLE [37] with Unipro UGENE software (version 34.0) [38] using default parameters. The phylogenetic tree was constructed by MrBayes, V. 3.27 using the GTR replacement model on the Unipro UGENE software platform (version 35.0).

Genome similarity of the strains ICIS 5, ICIS 9 and ICIS 53 and other *C. amycolatum* strains currently available at NCBI was determined by calculating the average nucleotide identity (ANI) and orthologous average nucleotide identity (OrthoANI) using OAT (version v. 0.93.1) software [39].

### 2.4. Nucleotide Sequence Accession Numbers

The annotated genome sequences were deposited in the GenBank database as sequencing project PRJNA339674 with accession numbers SSOR00000000, MTPT00000000 and MIFV00000000 for *C. amycolatum* ICIS 5, ICIS 9 and ICIS 53, respectively. The strains *C. amycolatum* ICIS 9 and ICIS 53 were deposited in the culture collection of the All-Russian Collection of Microorganisms at the G.K. Skryabin Institute of Biochemistry and Physiology of Microorganisms (Russian Academy of Sciences, Pushchino, Russia) under registration no. VKM Ac-2843D and VKM Ac-2844D, respectively.

## 3. Results

### 3.1. General Genome Features

As shown in Table 1, the draft genome of strain ICIS 5 was composed of 2,474,151 bp, with an N50 length of 164,886 bp, an L50 of 6, and a G + C content of 58.8%. The final assembled genome consisted of 115 contigs.

Genome annotation was performed using the National Center for Biotechnology Information (NCBI) Prokaryotic Genome Annotation Pipeline (PGAP) (http://www.ncbi.nlm.nih.gov/genome/annotation_prok (19 December 2021)), and 2195 coding sequences, including 2062 proteins (CDSs), 47 pseudogenes, complete rRNAs (3, 1 (5S, 16S) and 53 tRNAs, were identified. The identified coding proteins were classified into 26 functional categories based on COG classification (Appendix A). Of the 2062 protein-coding genes in ICIS 5, 1908 were assigned to COGs, and 154 genes were not assigned. The percentage of proteins with unknown function, including “Function unknown (S)” and “Not assigned (−)”, was 31.7%. Most genes belonged to the categories: Amino acid transport and metabolism (7.27% of CDS), Inorganic ion transport and metabolism (6.98% of CDS), Translation, ribosomal structure and biogenesis (6.84% of CDS), Replication, recombination and repair (6.3% of CDS) and Transcription (4.8% of CDS).

The genome of strain ICIS 9 was slightly larger than that of ICIS 5. It was composed of 2,587,830 bp, with an N50 length of 45,496 bp, an L50 of 18 and a G + C content of 58.6%. The final assembled genome consisted of 181 contigs. Genome annotation identified 2392 coding sequences, including 2277 proteins, 53 pseudogenes, complete rRNAs 1, 1, 1 (5S, 16S, 23S) and 53 tRNAs (Table 1). The identified coding proteins were classified into 26 functional categories based on COG classification (Appendix A). Of the 2277 protein-coding genes in ICIS 9, 2044 were assigned to COGs, and 233 genes were not assigned. The percentage of proteins with unknown function, including “Function unknown (S)” and “Not assigned (–)”, was 33.2%. Unlike ICIS 5, the distribution of identified coding proteins into categories based on COG classification was as follows: genes were mostly involved in the categories, Replication, recombination and repair (10.01% of CDS), Inorganic ion transport and metabolism (6.19% of CDS), Translation, ribosomal structure and biogenesis (6.19% of CDS), Amino acid transport and metabolism (6.05% of CDS) and Coenzyme transport and metabolism (4.61% of CDS).

The draft genome of strain ICIS 53 was composed of 2,460,257 bp, with an N50 length of 170,410 bp, an L50 of 4, and a G + C content of 59.0%. The final assembled genome consisted of 41 contigs. Genome annotation identified 2173 coding sequences, including 2076 proteins, 34 pseudogenes, 5, 1, and 1 complete rRNAs (5S, 16S, 23S) and 53 tRNAs (Table 1). The identified coding proteins were classified into 26 functional categories based on COG classification (Appendix A). Of the 2076 protein-coding genes in ICIS 53, 1884 were assigned to COGs, and 189 genes were not assigned. The percentage of proteins with unknown function, including “Function unknown (S)” and “Not assigned (–)”, was 33.9%. The identified coding proteins according to the COG classification were distributed as follows: Amino acid transport and metabolism (7.32% of CDS), Inorganic ion transport and metabolism (6.65% of CDS), Translation, ribosomal structure and biogenesis (6.55% of CDS), Replication, recombination and repair (4.82% of CDS) and Energy production and conversion (4.72% of CDS).

### 3.2. Phylogenetic Analysis

As shown in the phylogenetic tree constructed based on the 16S rRNA gene sequences, strains ICIS 5, ICIS 9, and ICIS 53 formed a common clade with nine strains of *C. amycolatum* from the NCBI database. The, *C. amycolatum* clade with a sister branch represented by *Corynebacterium xerosis* ATCC 373T was clearly separated from another clade containing all other species of *Corynebacterium* spp. (Figure 1). The data obtained are in good agreement with those described previously [40,41].

The similarity scores between ICIS 5, ICIS 9 and ICIS 53 and other *C. amycolatum* strains exceed 99% based on 16S rRNA gene phylogeny. The average nucleotide identity (ANI) between ICIS 5, ICIS 9 and ICIS 53 ranged from 96.79% to 97.87%, and the average nucleotide orthology (OrthoANI) ranged from 96.89% to 97.93% (Table 2).

ANI values between our three strains and a type SK 46 strain of *C. amycolatum* varied from 94.93% to 95.29%, and orthoANI values varied from 95.05% to 95.37% and were over the species boundary value (ANI > 95–96%, orthoANI > 95–96%) [39,42]. Relative to other *C. amycolatum* strains, ANI values ranged from 94.26% to 98.02%, OrthoANI values ranged from 94.3% to 98.1%.

### 3.3. Genome Annotation of ICIS 5, ICIS 9 and ICIS 53

#### 3.3.1. Genes and Properties Allowing, *C. amycolatum* to Survive in the Vaginal Ecosystem

The vaginal ecosystem is an aggressive environment for most microorganisms. The acidic environment in the vagina creates a natural filter; as a result, most pathogens and opportunistic microbes die. In order to survive and successfully colonise this ecosystem, microorganisms of the genus *Corynebacterium* spp. must have evolved mechanisms of adaptation. In the studied genomes, we identified a large number of genes encoding proteins involved in stress response. These stresses included pH, temperature, osmotic pressure, nitrosative and oxidative stress. The detailed analysis of genes coding for proteins involved in stress response in the genomes of ICIS 5, ICIS 9 and ICIS 53 is shown in Table 3.

ICIS 5, ICIS 9 and ICIS 53 contain 7 genes that encode F0F1-ATPase. Membrane-bound ATP synthases (F0F1-ATPases) of bacteria serve two important physiological functions. The enzyme catalyses the synthesis of ATP from ADP and inorganic phosphate utilizing the energy of an electrochemical ion gradient. On the other hand, under conditions of low driving force, ATP synthases function as ATPases, thereby generating a transmembrane ion gradient at the expense of ATP hydrolysis [43]. Such activity protects cells from damage induced by an acidic environment; 3 genes encoding Na^+^/H^+^ antiporters are membrane proteins that play a major role in pH and Na^+^ homeostasis of cells [44]. The analysis of the genomes revealed the presence of genes that encode L-lactate dehydrogenase. This enzyme catalyses the conversion of lactate to pyruvate with the formation of NADH. As a result, it restored the NAD/NADH balance and subsequently increased ATP production. The concomitant surplus of ATP is used to drive the F0F1-ATPases, resulting in enhanced acid tolerance in bacteria [45]. Furthermore, the ICIS 5, ICIS 9 and ICIS 53 genomes encode glucose-6-phosphate isomerase, GTP pyrophosphokinase, pyruvate kinase, ATP-dependent Clp protease ATP-binding subunit, which are proteins involved in the acid resistance of various bacteria [46,47]. We also identified a number of genes related to temperature stress. A cluster of heat shock proteins was identified, hrcA-grpE-dnaK-dnaJ and chaperonin system GroEL-GroES, which are present in all kingdoms of life and rescue proteins from improper folding and aggregation upon internal and external stress conditions, including high temperatures and pressures [48,49].

A large number of genes associated with oxidative stress were identified in the studied genomes. They can play a crucial significance for survival and adaptation of the bacteria in the vaginal niche. The genomes contain genes that encode catalase, thiol peroxidase and glutathione peroxidase, which are antioxidant protective enzymes capable to detoxify reactive oxygen species [50,51,52]. The genomes also harbour genes encoding superoxide dismutase (SOD) and the complete thioredoxin system. It is known that SOD and thioredoxin (Trx) systems are key antioxidant systems in cellular protection against oxidative stress conditions [53,54,55,56]. In addition, genes nrdH and MSH encoding glutaredoxin and mycothiol, respectively, were identified. Similar to glutathione, mycothiol is one of key metabolites providing protection of bacteria from oxidative stress, as well as detoxication of xenobiotics [57,58]. Recently, significance of MSH has been shown for resistance of *Corynebacterium glutamicum* to antibiotics, alkylating agents, ethanol and heavy metals [59,60]. The genomes of strains ICIS 5, ICIS 9 and ICIS 53 contained genes encoding four enzymatic steps of mycothiol biosynthesis: production of GlcNAc-Ins-P using D-inositol-3-phosphate glycosyltransferase (MshA), deacetylation using N-acetyl-1-D-myo-inositol-2-amino-2-deoxy-alpha-D-glucopyranoside deacetylase (MshB) to form GlcN-Ins, binding to cysteine via cysteine-1-D-myo-inosityl-2-amino-2-deoxy-alpha-D-glucopyranoside ligase (MshC), and acetylation of mycothiol synthase (MshD) to give MSH.

#### 3.3.2. Biologically Active Secondary Metabolite-Related Genes

In order to survive and successfully colonise the vaginal biotope, nonpathogenic corynebacteria must have the ability to produce secondary metabolites with antimicrobial activity and determine their competitive advantage [61]. In the studied genomes, we found the presence of gene clusters potentially involved in the biosynthesis of secondary metabolites. Gene clusters were predicted for T3pks (type III polyketide synthases), Nrps (nonribosomal peptide), Nrps-like and terpene. Each of the three genomes contained one T3pks gene cluster, which was associated with the biosynthesis of polyketides. Polyketides are natural metabolites that comprise the basic chemical structure of various anticancer, antifungal and anticholesteremic agents, antibiotics, parasiticides and immunomodulators [62,63]. These T3pks gene clusters encoded the biosynthesis of merochlorin A–D-like compounds (Appendix A). Merochlorins A–D, cyclic meroterpenoid antibiotics, were first described in the marine bacterium Streptomyces sp. strain CNH-189 [64]. The genomes of strains ICIS 5 and ICIS 53 contained one Nrps gene cluster, which was associated with the biosynthesis of phthoxazolin-like compounds (Appendix A). Phthoxazolin, an oxazole-containing polyketide, has a broad spectrum of anti-oomycete activity and herbicidal activity [65]. Additionally, each genome contained one terpene gene cluster and an Nrps-like gene cluster, but substances were not identified. Terpenes or isoprenoids are the largest and structurally most diverse class of secondary metabolites. Terpenes are involved in a wide range of vital biological functions, including electron transport, cellular respiration, photosynthesis, membrane biosynthesis, signalling and growth regulation [66]. In accordance with their structural diversity, the functions of terpenoids range from mediating symbiotic or antagonistic interactions between organisms to electron transfer, protein prenylation, or contribution to membrane fluidity [67]. In addition, an increasing number of terpenes have been utilised for pharmaceuticals [68,69,70]. We checked the “Terpenoid backbone biosynthesis (map00900)” pathway in the genomes of strains ICIS 5, ICIS 9 and ICIS 53 and identified six key enzymes distributed in the mevalonate (MVA) pathway. The obtained data confirm the previously described MVA pathway for terpenoid backbone biosynthesis in *C. amycolatum* [71]. The core enzymes involved in the MVA pathway are listed in Table 4.

All enzymes are encoded by a single gene, except isoprenyl transferase (undecaprenyl diphosphate synthase), which is encoded by two gene copies. Isoprenyl transferase catalyses the condensation of isopentenyl diphosphate (IPP) with allylic pyrophosphates, generating different types of terpenoids [72]. Farnesyl-diphosphate farnesyltransferase (squalene synthase) is a precursor of steroids, cholesterol, sesquiterpenes, farnesylated proteins, heme and vitamin K12 [73].

#### 3.3.3. Bacteriocin-Related Genes

Bacteriocins are antimicrobial peptides ribosomally produced in bacteria, either processed or not by additional post-translational modification (PTM) enzymes, and exported to the extracellular medium [74,75]. Of the three genomes analysed, only the genome of strain ICIS 9 had one area of interest (AOI) that included genes encoding a bacteriocin of the class Sactipeptide (Figure 2 and Appendix A).

Sactipeptides are a new class of synthesized in ribosomes and post-translationally modified peptides (RiPPs). Sactipeptides are known as antibiotics with narrow spectrum capable to inhibit *Clostridia* and some human multidrug-resistant bacterial pathogens [76]. The revealed features allow to consider sactipeptides promising scaffolds for the creation of new antibiotics [77]. The presence of an AOI encoding sactipeptide in the genome of strain ICIS 9 suggests that this strain may produce the sactipeptide. However, this assumed feature should be checked further through isolation and characteristics of this peptide.

#### 3.3.4. H_2_O_2_-Related Genes

H_2_O_2_ is one of the key factors in maintaining vaginal biotope homeostasis [78]. In the studied genomes, we identified a number of genes encoding the production of hydrogen peroxide: cytochrome d ubiquinol oxidase subunit I (locus_tag: E7L51_RS08995, BXT90_RS09475, BGC22_RS06540 and subunit II (E7L51_RS08990, BXT90_RS09470, BGC22_RS06535), glutathione peroxidase family protein (E7L51_RS08875, BXT90_RS09805, BGC22_RS06430), pyridoxamine 5′-phosphate oxidase (E7L51_RS09830, BXT90_RS11320, BGC22_RS08135), fumarate reductase (E7L51_RS09150, BXT90_RS09225, BGC22_RS06825), and NADH-flavin reductase family protein (E7L51_RS07920, BXT90_RS06560, BGC22_RS02940). As shown in the Lactobacillus acidophilus group [79], these enzymes are directly involved in the production of hydrogen peroxide, along with well-known enzymes such as NADH oxidase or lactate oxidase.

#### 3.3.5. Nutrient Synthesis (Vitamins and Essential Amino Acids)-Related Genes

Microorganisms that colonize various biotopes of the human body form multi-species communities and represent a kind of “organ”, which, in turn, affects the functioning of all organs and systems that play an important role in maintaining the health of the host. Using intestinal microbiota as an example, commensal bacteria have been shown to be important sources of vitamins and amino acids. In addition to their nutritional/physiological properties, many of these vitamins are also involved in the development and functioning of host immune cells, as there is a direct link between biosynthetic biosynthesis intermediates derived from commensal bacteria and immune cells that directly recognise them [80,81]. The biosynthesis of vitamins and essential amino acids by probiotic strains has recently been an important aspect in the development of probiotic products and pharmaceuticals [82,83]. The genomes of strains ICIS 5, ICIS 9 and ICIS 53 contain functionally active biosynthetic gene clusters that encode all the enzymes required for the synthesis of B vitamins such as B2 (riboflavin), B6 (pyridoxin), B7 (biotin), B9 (folate) and B12 (cobalamin) (Table 5), and essential amino acids such as histidine, arginine, methionine, threonine, lysine, leucine and tryptophan (Table 6).

#### 3.3.6. Antibiotic Resistance- and Virulence-Related Genes

We searched the genomes of vaginal isolates of *C. amycolatum* strains for antibiotic resistance genes and found genes encoding resistance to antibiotics in only two strains, ICIS 5 and ICIS 9. The genome of strain ICIS 5 contained genes encoding resistance to chloramphenicol and aminoglycosides (Appendix A). The genome of strain ICIS 9 contained genes encoding resistance to macrolides, lincosamide, streptogramin, tetracycline, chloramphenicol and aminoglycosides (Appendix A). The results confirmed the antibiotic resistance profile of these strains, which was determined earlier with a disc susceptibility assay [84]. VFDB software was used to predict virulence factors in ICIS 5, ICIS 9 and ICIS 53. VFDB predicted 27 virulence factors in ICIS 5 and ICIS 9 and 23 virulence factors in ICIS 53 (Appendix A). The virulence genes of ICIS 5, ICIS 9 and ICIS 53 can be classified into ten categories: adherence, iron uptake, regulation, amino acid and purine metabolism, antiphagocytosis, cell surface components, immune evasion, lipid and fatty acid metabolism, protease and secretion system. However, all identified genes were not true virulence factors; only the structural and functional characteristics of microorganisms of the genus *Corynebacterium* were determined, as well as adaptation to this ecological niche [85,86,87,88,89]. The true virulence genes, such as toxin-related genes (diphtheria toxin and phospholipase D) and haemolysin-related genes characteristic of well-known pathogenic corynebacteria, were not identified.

#### 3.3.7. Phage Defense Systems

CRISPR-Cas modules are adaptive immune systems that are present in most archaea and many bacteria and provide sequence-specific protection against foreign DNA or, in some cases, RNA [90]. Of the three analysed genomes, CRISPR-associated sequence (Cas) systems were identified exclusively in the genome of strain ICIS 5. Type I-E CRISPR consists of 7 cas genes: cas1e (locus_tag: E7L51_RS06945), cas2e (E7L51_RS06950), cas3 E7L51_RS06915, Cse4 (E7L51_RS06930), cas5e (E7L51_RS06935), cas6e (E7L51_RS06940) and cas7e (E7L51_RS06930). The abortive infection (Abi) system is another property of phage resistance that can target different phases of phage development [91]. Abortive infection family proteins were identified in the genomes of strains ICIS 5 (locus_tag: E7L51_RS07100), ICIS 9 (BXT90_RS06835) and ICIS 53 (BGC22_RS10885). The presence of such phage defence systems in the studied strains probably reflects the exposure of these strains to phages in the vagina [92].

## 4. Conclusions

We presented a comparative study of draft genomes for three *C. amycolatum* vaginal strains based on the annotation and analysis of genes associated with the physiological functions and adaptation of these microorganisms under the specific conditions of vaginal microbiocenosis. The presence of resistance genes against acid and oxidative stress, which are specific features of the vaginal biotope, has been established. The genes responsible for the synthesis of essential amino acids and vitamins have been identified, demonstrating the involvement of the corynebacteria in host metabolism. The presence of genes associated with H_2_O_2_ production and the absence of true virulence genes allows us to consider the studied strains as potential probiotics capable of preventing the growth of pathogens and supporting colonisation resistance in the vaginal biotope. Antibiotic resistance genes revealed in the strains can provide corynebacteria the ability to colonise the vaginal biotope under antibiotic treatment. In addition, the studied strains are biotechnologically promising based on the identified genes for the synthesis of secondary metabolites of polyketides (merochlorins A–D), terpenes and sactipeptides. The analysis of *C. amycolatum* genomes revealed common genes for all three strains; for example, genes encoding adaptation and survival in the vaginal environment, as well as unique genes determining strain specificity. The presence of an AOI encoding sactipeptide in the genome of strain ICIS 9 suggests that this strain may produce the sactipeptide. However, this assumed feature should be checked further through isolation and characteristics of this peptide.

## Figures and Tables

**Figure 1 microorganisms-10-00249-f001:**
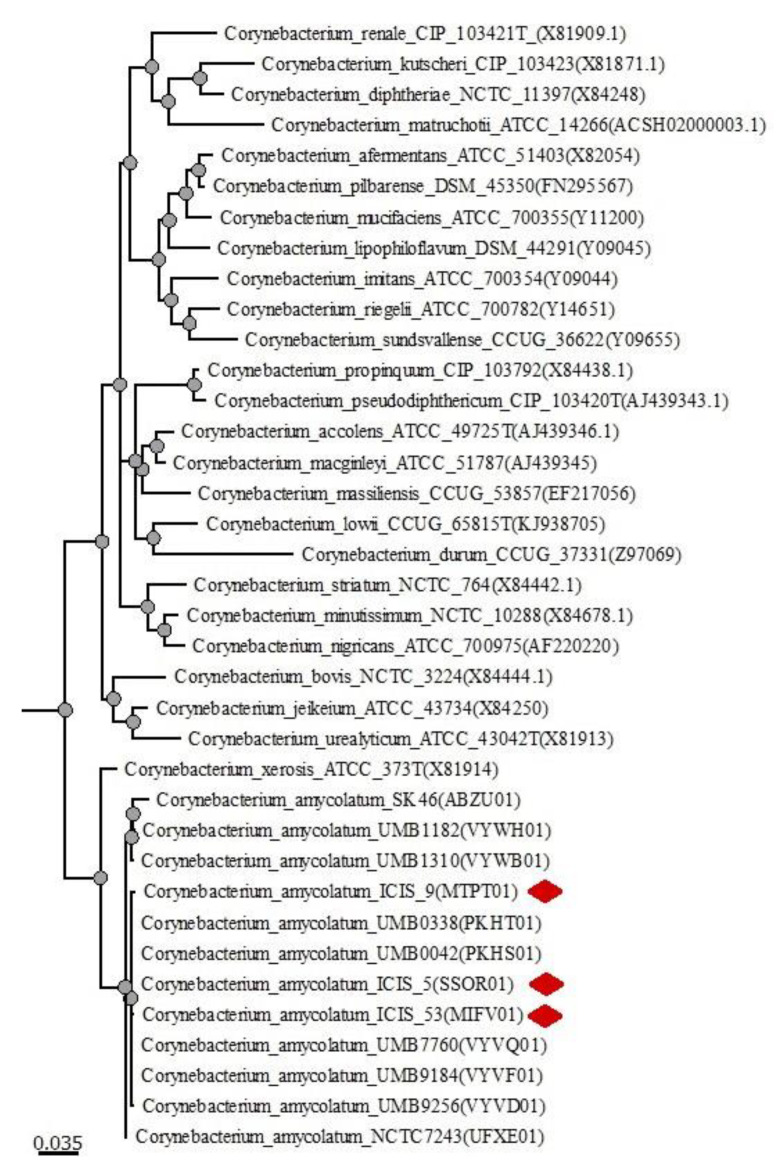
Phylogenetic tree highlighting the position of *C. amycolatum* ICIS 5, ICIS 9, and ICIS 53 (denoted with red diamonds) relative to the 28 type strains of *Corynebacterium* spp. most often isolated from human clinical samples and 2 strains of *Euzebya* spp. as an out group. The phylogenetic tree was constructed by MrBayes, V. 3.27 using the GTR replacement model on the Unipro UGENE software platform (version 35.0). Corresponding NCBI accession numbers are shown in parentheses.

**Figure 2 microorganisms-10-00249-f002:**
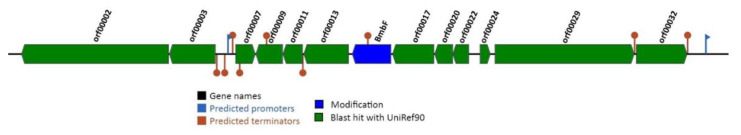
Sactipeptide BmbF encoded by gene *bmbF* (orf00016) predicted in ICIS 9 genome with BAGEL4.

**Table 1 microorganisms-10-00249-t001:** The characteristics of the assembly and genomes of *Corynebacterium amycolatum* strains.

Statistics	ICIS 5	ICIS 9	ICIS 53
Assembly			
Number of contigs	115	181	41
N50	164,886	45,496	170,644
L50	6	18	4
Depth of coverage	278	22	100
Draft genome sequences			
Genome size (b.p.)	2,474,151	2,587,830	2,460,257
GC contents (%)	58.80	58.60	59.00
Genes (total)	2195	2392	2173
CDSs (total)	2109	2330	2110
Genes (coding)	2062	2277	2076
CDSs (with protein)	2062	2277	2076
rRNAs (5S, 16S, 23S)	30 (4, 20, 6)	6 (4, 1, 1)	7 (5, 1, 1)
complete rRNAs	3, 1 (5S, 16S)	1, 1, 1 (5S, 16S, 23S)	5, 1, 1 (5S, 16S, 23S)
tRNAs	53	53	53
Pseudo Genes (total)	47	53	34

**Table 2 microorganisms-10-00249-t002:** Heatmap showing relative average nucelotide identity (ANI) and average nucleotide orthology (OrthoANI) between *C. amycolatum* species.

	(OrthoANI)											
	ICIS5	ICIS9	ICIS53	NCTC7243	SK46	UMB0042	UMB0338	UMB1182	UMB7760	UMB9184	UMB1310	UMB9256
ICIS5		96.87	97.93	94.97	95.38	97.95	97.51	94.96	97.69	97.57	95.13	97.40
ICIS9	96.79		96.90	95.71	95.05	96.83	96.79	95.42	96.83	96.76	95.55	96.66
ICIS53	97.87	96.84		94.77	95.37	98.07	97.74	94.95	97.82	97.83	94.96	97.47
NCTC7243	94.90	95.66	94.74		94.30	94.77	95.13	96.87	95.03	95.08	96.77	95.21
SK46	95.20	94.93	95.29	94.22		95.35	95.09	94.42	95.16	95.17	94.43	95.12
UMB0042	97.85	96.78	98.05	94.67	95.29		97.54	94.73	97.66	97.55	94.88	97.26
UMB0338	97.48	96.71	97.67	95.02	95.05	97.55		95.04	97.89	97.85	95.10	97.66
UMB1182	94.83	95.37	94.75	96.83	94.26	94.62	94.90		95.06	94.96	98.70	95.20
UMB7760	97.69	96.74	97.82	95.00	95.12	97.66	97.81	94.93		98.10	95.12	97.98
UMB9184	97.58	96.75	97.79	94.97	95.11	97.60	97.80	94.90	98.03		95.07	97.68
UMB1310	95.02	95.50	94.85	96.68	94.28	94.82	95.07	98.64	95.06	95.03		95.23
UMB9256	97.33	96.58	97.47	95.18	95.06	97.30	97.58	95.08	97.95	97.67	95.18	
	**(Original ANI)**											

**Table 3 microorganisms-10-00249-t003:** Genes coding for proteins involved in stress response detected in vaginal isolates *Corynebacterium amycolatum* strains.

Stresses	Gene	Product	ICIS 5 Locus_tag	ICIS 9 Locus_tag	ICIS 53 Locus_tag
pH	*atpB*	F0F1 ATP synthase subunit A	E7L51_RS04555	BXT90_RS01980	BGC22_RS01625
	–	F0F1 ATP synthase subunit B	E7L51_RS04545	BXT90_RS01970	BGC22_RS01615
		F0F1 ATP synthase subunit C			
	*–*	F0F1 ATP synthase subunit alpha	E7L51_RS04535	BXT90_RS01960	BGC22_RS01605
	*atpD*	F0F1 ATP synthase subunit beta	E7L51_RS04525	BXT90_RS01950	BGC22_RS01595
	*–*	F0F1 ATP synthase subunit gamma	E7L51_RS04530	BXT90_RS01955	BGC22_RS01600
	*–*	F0F1 ATP synthase subunit delta	E7L51_RS04540	BXT90_RS01965	BGC22_RS01610
	*–*	F0F1 ATP synthase subunit epsilon	E7L51_RS04520	BXT90_RS01945	BGC22_RS01590
		F0F1 ATP synthase protein I			
	*–*	L-lactate dehydrogenase	E7L51_RS05660, E7L51_RS02260	BXT90_RS06700, BXT90_RS07585	BGC22_RS01270, BGC22_RS10270
	*–*	Glucose-6-phosphate isomerase	E7L51_RS06505	BXT90_RS00390	BGC22_RS07690
	*–*	GTP pyrophosphokinase	E7L51_RS00625	BXT90_RS01675	BGC22_RS04770
	*pyk*	Pyruvate kinase	E7L51_RS00810, E7L51_RS02265	BXT90_RS02290, BXT90_RS06705	BGC22_RS04585, BGC22_RS10275
	*clpX*	ATP-dependent Clp protease ATP-binding subunit	E7L51_RS09205	BXT90_RS08190, BXT90_RS06490	BGC22_RS03025, BGC22_RS06215
	–	Na+/H+ antiporter subunit A	E7L51_RS03140	BXT90_RS04255	BGC22_RS02670
	–	Na+/H+ antiporter subunit D	E7L51_RS03130	BXT90_RS04265	BGC22_RS02680
	–	Na+/H+ antiporter subunit E	E7L51_RS03125	BXT90_RS04270	BGC22_RS02685
Temperature	*hrcA*	Heat-inducible transcriptional repressor	E7L51_RS03385	BXT90_RS03730	BGC22_RS05570
	*grpE*	Heat shock protein GrpE	E7L51_RS02840	BXT90_RS09385	BGC22_RS02385
	*dnaK*	Heat shock protein DnaK	E7L51_RS02835	BXT90_RS09390	BGC22_RS02390
	*dnaJ*	Heat shock protein DnaJ	E7L51_RS02845, E7L51_RS03380, E7L51_RS06605	BXT90_RS03725, BXT90_RS09380	BGC22_RS02380, BGC22_RS07585, BGC22_RS05565
	*–*	Molecular chaperone GroES	E7L51_RS06730	BXT90_RS00160	BGC22_RS07460
	*groL*	Molecular chaperone GroEL	E7L51_RS03170, E7L51_RS06725	BXT90_RS00165, BXT90_RS06205	BGC22_RS02645, BGC22_RS07465
Osmotic stress	*betA*	Choline dehydrogenase (EC 1.1.99.1)	E7L51_RS02520	BXT90_RS09090	BGC22_RS06865
Nitrosative stress	–	Nitrate reductase	E7L51_RS05075	BXT90_RS02975	BGC22_RS00655
Oxidative stress	*–*	Catalase (EC 1.11.1.6)	E7L51_RS10255	BXT90_RS07525	BGC22_RS03550
	*–*	Thiol peroxidase	E7L51_RS01250	BXT90_RS05315	BGC22_RS08510
	*trxA*	Thioredoxin	E7L51_RS08530, E7L51_RS08670	BXT90_RS04815, BXT90_RS10200	BGC22_RS02030, BGC22_RS02165
	*trxB*	Thioredoxin-disulfide reductase	E7L51_RS08665	BXT90_RS04820	BGC22_RS02160
	*–*	Thioredoxin-dependent thiol peroxidase	E7L51_RS10400	BXT90_RS08035	BGC22_RS04010
	*–*	Thioredoxin domain-containing protein	E7L51_RS09080, E7L51_RS10075	BXT90_RS09275, BXT90_RS07735	BGC22_RS03780, BGC22_RS08125
	*–*	Glutathione peroxidase	E7L51_RS08875	BXT90_RS09805	BGC22_RS06430
	*–*	Hydrogen peroxide-inducible genes activator	E7L51_RS05005	BXT90_RS03045	BGC22_RS00585
	*–*	Superoxide dismutase	E7L51_RS02250	BXT90_RS06690	BGC22_RS10260
	*mshA*	D-inositol-3-phosphate glycosyltransferase	E7L51_RS05705	BXT90_RS09175	BGC22_RS06775
	*mshB*	N-acetyl-1-D-myo-inositol-2-amino-2-deoxy-alpha-D-glucopyranoside deacetylase	E7L51_RS04900	BXT90_RS09615	BGC22_RS01970
	*mshC*	Cysteine--1-D-myo-inosityl 2-amino-2-deoxy-alpha-D-glucopyranoside ligase	E7L51_RS01445	BXT90_RS02610	BGC22_RS08710
	*mshD*	Mycothiol synthase	E7L51_RS08935	BXT90_RS09425	BGC22_RS06490
	*mca*	Mycothiol conjugate amidase	E7L51_RS03790	BXT90_RS06270	BGC22_RS05930
	*mtr*	Mycothione reductase	E7L51_RS05350	BXT90_RS05885	BGC22_RS00925
	*nrdH*	Gutaredoxin-like protein	E7L51_RS09855	BXT90_RS10670	BGC22_RS03865

**Table 4 microorganisms-10-00249-t004:** Genes coding for proteins involved in terpenoid backbone biosynthesis in *Corynebacterium amycolatum* strains.

Product/EC No.	ICIS 5 Locus_tag	ICIS 9 Locus_tag	ICIS 53 Locus_tag
acetyl-CoA C-acetyltransferase (EC:2.3.1.9)	E7L51_RS02615 E7L51_RS08850	BXT90_RS00480	BGC22_RS02560 BGC22_RS06405 BGC22_RS07780
hydroxymethylglutaryl-CoA synthase (EC:2.3.3.10)	E7L51_RS06810	BXT90_RS00080	BGC22_RS07380
hydroxymethylglutaryl-CoA reductase (NADPH) (EC:1.1.1.34)	E7L51_RS06815	BXT90_RS00075	BGC22_RS07375
mevalonate kinase (EC:2.7.1.36)	E7L51_RS06830	BXT90_RS00060	BGC22_RS07360
phosphomevalonate kinase (EC:2.7.4.2)	E7L51_RS06820	BXT90_RS00070	BGC22_RS07370
diphosphomevalonate decarboxylase (EC:4.1.1.33)	E7L51_RS06825	BXT90_RS00065	BGC22_RS07365
isoprenyl transferase (undecaprenyl diphosphate synthase) uppS (EC:2.5.1-)	E7L51_RS03345 E7L51_RS03775	BXT90_RS03690 BXT90_RS06255	BGC22_RS05530 BGC22_RS05915
isopentenyl-diphosphate Delta-isomerase (EC:5.3.3.2)	E7L51_RS06835	BXT90_RS00055	BGC22_RS07355
polyprenyl synthetase family protein	E7L51_RS05960	BXT90_RS05620	BGC22_RS04055
geranylgeranyl pyrophosphate synthase (EC:2.5.1.1 2.5.1.10 2.5.1.29)	E7L51_RS00410	BXT90_RS01380	BGC22_RS04980
farnesyl-diphosphate farnesyltransferase (EC:2.5.1.21)	E7L51_RS00395	BXT90_RS01365	BGC22_RS04995

**Table 5 microorganisms-10-00249-t005:** Vitamin biosynthetic proteins detected in *Corynebacterium amycolatum* ICIS 5, ICIS 9 and ICIS 53.

Vitamin	Biosynthesis Protein/Gene/EC No.
Biotin	Transmembrane component BioN of energizing module of biotin ECF transporter Predicted biotin repressor from TetR family Substrate-specific component BioY of biotin ECF transporter Adenosylmethionine-8-amino-7-oxononanoate aminotransferase (EC 2.6.1.62) 8-amino-7-oxononanoate synthase (EC 2.3.1.47) Dethiobiotin synthetase (EC 6.3.3.3) Biotin synthase (EC 2.8.1.6) Long-chain-fatty-acid--CoA ligase (EC 6.2.1.3) Biotin synthesis protein BioC 3-ketoacyl-CoA thiolase (EC 2.3.1.16) ATPase component BioM of energizing module of biotin ECF transporter
Cobalamin	Cobalt-precorrin-6x reductase (EC 1.3.1.54) Cobalamin biosynthesis protein BluB L-threonine 3-O-phosphate decarboxylase (EC 4.1.1.81) Adenosylcobinamide-phosphate guanylyltransferase (EC 2.7.7.62) Cobalt-precorrin-8x methylmutase (EC 5.4.1.2) Cobalt-precorrin-2 C20-methyltransferase (EC 2.1.1.130) Cobyric acid synthase (EC 6.3.5.10) Cobalt-precorrin-4 C11-methyltransferase (EC 2.1.1.133) Cobalt-precorrin-3b C17-methyltransferase Nicotinate-nucleotide--dimethylbenzimidazole phosphoribosyltransferase (EC 2.4.2.21) Adenosylcobinamide-phosphate synthase (EC 6.3.1.10) Cob(I)alamin adenosyltransferase (EC 2.5.1.17) Cobyrinic acid A,C-diamide synthase
Riboflavin	FMN adenylyltransferase (EC 2.7.7.2) hypothetical protein YebC C-terminal domain of CinA type S 6,7-dimethyl-8-ribityllumazine synthase (EC 2.5.1.78) Riboflavin transporter PnuX 5-amino-6-(5-phosphoribosylamino)uracil reductase (EC 1.1.1.193) tRNA pseudouridine synthase B (EC 4.2.1.70) Riboflavin kinase (EC 2.7.1.26) GTP cyclohydrolase II (EC 3.5.4.25) Diaminohydroxyphosphoribosylaminopyrimidine deaminase (EC 3.5.4.26) 3,4-dihydroxy-2-butanone 4-phosphate synthase (EC 4.1.99.12) FMN adenylyltransferase (EC 2.7.7.2) Riboflavin synthase eubacterial/eukaryotic (EC 2.5.1.9)
Pyridoxine	Pyridoxine biosynthesis glutamine amidotransferase, synthase subunit (EC 2.4.2.-) D-3-phosphoglycerate dehydrogenase (EC 1.1.1.95) Pyridoxal kinase (EC 2.7.1.35) Phosphoserine aminotransferase (EC 2.6.1.52) NAD-dependent glyceraldehyde-3-phosphate dehydrogenase (EC 1.2.1.12)
Folate	Dihydropteroate synthase (EC 2.5.1.15) tRNA(Ile)-lysidine synthetase (EC 6.3.4.19) Aspartate 1-decarboxylase (EC 4.1.1.11) Cell division protein FtsH (EC 3.4.24.-) GTP cyclohydrolase I (EC 3.5.4.16) type 1 Pantoate--beta-alanine ligase (EC 6.3.2.1) 2-amino-4-hydroxy-6-hydroxymethyldihydropteridine pyrophosphokinase (EC 2.7.6.3) Dihydroneopterin aldolase (EC 4.1.2.25) 5-formyltetrahydrofolate cyclo-ligase (EC 6.3.3.2) Dihydrofolate reductase (EC 1.5.1.3) Thymidylate synthase thyX (EC 2.1.1.-) Dihydrofolate synthase (EC 6.3.2.12) Para-aminobenzoate synthase, aminase component (EC 2.6.1.85)

**Table 6 microorganisms-10-00249-t006:** Amino acid biosynthetic proteins detected in *Corynebacterium amycolatum* ICIS 5, ICIS 9 and ICIS 53.

Amino Acid	Biosynthesis Protein/Gene/EC No.
Histidine	Phosphoribosylformimino-5-aminoimidazole carboxamide ribotide isomerase (EC 5.3.1.16) Phosphoribosyl-ATP pyrophosphatase (EC 3.6.1.31) Imidazole glycerol phosphate synthase amidotransferase subunit (EC 2.4.2.-) Histidinol-phosphatase [alternative form] (EC 3.1.3.15) Histidinol-phosphate aminotransferase (EC 2.6.1.9) Imidazoleglycerol-phosphate dehydratase (EC 4.2.1.19) Imidazole glycerol phosphate synthase cyclase subunit (EC 4.1.3.-) Phosphoribosyl-AMP cyclohydrolase (EC 3.5.4.19) ATP phosphoribosyltransferase (EC 2.4.2.17) Histidinol dehydrogenase (EC 1.1.1.23)
Arginine	N-succinyl-L,L-diaminopimelate desuccinylase (EC 3.5.1.18) Glutamate N-acetyltransferase (EC 2.3.1.35) Acetylglutamate kinase (EC 2.7.2.8) Arginine pathway regulatory protein ArgR Argininosuccinate lyase (EC 4.3.2.1) N-acetylglutamate synthase (EC 2.3.1.1) Argininosuccinate synthase (EC 6.3.4.5) N-acetyl-gamma-glutamyl-phosphate reductase (EC 1.2.1.38) Ornithine carbamoyltransferase (EC 2.1.3.3) Acetylornithine aminotransferase (EC 2.6.1.11)
Methionine	Methionine ABC transporter ATP-binding protein S-adenosylmethionine synthetase (EC 2.5.1.6) O-succinylhomoserine sulfhydrylase (EC 2.5.1.48) Serine acetyltransferase (EC 2.3.1.30) Homoserine kinase (EC 2.7.1.39) 5-methyltetrahydropteroyltriglutamate--homocysteine methyltransferase (EC 2.1.1.14) 5-methyltetrahydrofolate--homocysteine methyltransferase (EC 2.1.1.13) Cystathionine beta-lyase, type II (EC 4.4.1.8) Cysteine synthase (EC 2.5.1.47) O-acetylhomoserine sulfhydrylase (EC 2.5.1.49) 5,10-methylenetetrahydrofolate reductase (EC 1.5.1.20) Homoserine dehydrogenase (EC 1.1.1.3) Methionine ABC transporter substrate-binding protein Homoserine O-acetyltransferase (EC 2.3.1.31) Methionine ABC transporter permease protein Adenosylhomocysteinase (EC 3.3.1.1)
Threonine	Homoserine dehydrogenase (EC 1.1.1.3) Aspartate-semialdehyde dehydrogenase (EC 1.2.1.11) Aspartate aminotransferase (EC 2.6.1.1) Threonine synthase (EC 4.2.3.1) Homoserine kinase (EC 2.7.1.39) Aspartokinase (EC 2.7.2.4)
Lysine	N-succinyl-L,L-diaminopimelate desuccinylase (EC 3.5.1.18) N-acetyl-L,L-diaminopimelate deacetylase (EC 3.5.1.47) 2,3,4,5-tetrahydropyridine-2,6-dicarboxylate N-succinyltransferase (EC 2.3.1.117) Diaminopimelate epimerase (EC 5.1.1.7) Aspartate-semialdehyde dehydrogenase (EC 1.2.1.11) 4-hydroxy-tetrahydrodipicolinate synthase (EC 4.3.3.7) Meso-diaminopimelate D-dehydrogenase (EC 1.4.1.16) Diaminopimelate decarboxylase (EC 4.1.1.20) N-succinyl-L,L-diaminopimelate aminotransferase alternative (EC 2.6.1.17) 2,3,4,5-tetrahydropyridine-2,6-dicarboxylate N-acetyltransferase (EC 2.3.1.89) 4-hydroxy-tetrahydrodipicolinate reductase (EC 1.17.1.8) Aspartokinase (EC 2.7.2.4)
Leucine	3-isopropylmalate dehydratase small subunit (EC 4.2.1.33) 3-isopropylmalate dehydratase large subunit (EC 4.2.1.33) 2-isopropylmalate synthase (EC 2.3.3.13) Branched-chain amino acid aminotransferase (EC 2.6.1.42) 3-isopropylmalate dehydrogenase (EC 1.1.1.85)
Tryptophan	Anthranilate synthase, amidotransferase component (EC 4.1.3.27) Aminodeoxychorismate lyase (EC 4.1.3.38) Tryptophan-associated membrane protein Tryptophan synthase alpha chain (EC 4.2.1.20) Anthranilate phosphoribosyltransferase (EC 2.4.2.18) Tryptophan synthase beta chain (EC 4.2.1.20) Acting phosphoribosylanthranilate isomerase (EC 5.3.1.24) Indole-3-glycerol phosphate synthase (EC 4.1.1.48) Anthranilate synthase, aminase component (EC 4.1.3.27) Para-aminobenzoate synthase, aminase component (EC 2.6.1.85) Para-aminobenzoate synthase, amidotransferase component (EC 2.6.1.85)

## Data Availability

All data presented in this study are available in the article.

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
