# Peer review of "Genome Characterization and Probiotic Potential of Corynebacterium amycolatum Human Vaginal Isolates"

_microorganisms, 2022, doi:10.3390/microorganisms10020249_

Round 1

Reviewer 1 Report

Review of the article: “Genome characterization and probiotic potential of Corynebacterium amycolatum human vaginal isolates”

Submission ID - microorganisms-1539656

I have read the manuscript with great interest. In my opinion the authors performed fully professional bioinformatics analysis of genomes of three strains of C. amycolatum strains isolated from vaginal biotope of healthy  women. I do not have any critical remarks about methodology – as I wrote above it is a really professional analysis. Some results are promising (from practical point of view – possibilities of application of isolated strain/strains as probiotics) and interesting (from scientific point of view). The strain that is (probably) able to produce bacteriocins seems to particularly interesting. Identification of some other genes is also very important – e.g. genes responsible for production of hydrogen peroxide and antibiotic resistance. Moreover, the authors revealed that the isolated bacteria do not produce “important” virulence factors, which is also very important.  However, I have also some critical remarks about the study. First of all I regret that the authors did not perform any experiments that could finally confirm antibiotic resistance or production of bacteriocins. I believe that the authors are going to perform these analysis in the near future. I also would be grateful for additional comment about possibilities of application of isolated strains – the strains were isolated from vaginal environment and I understand that they are able to survive under conditions that are typical for this biotope, but what about skin or gastrointestinal tract (are they able to survive under these conditions) – some probiotics are important for protection our skin and gastrointestinal tract against development  of pathogenic bacteria (e.g. during antibiotic therapies). In introduction the authors have written: “In addition, a large group of nondiphtheria corynebacteria is part of the resident microflora of human skin and mucous membranes and most often stands out from clinical samples”.

Less important comments:

In some brackets the authors have written “Figura” instead of Figure. The authors should use italic for presenting names of bacteria – please go through the whole text and write the names of microorganisms according to generally accepted rules.

In introduction, the authors have written:” It was shown that certain types of nondiphtheria  corynebacteria produce various bacteriocins, bacteriocin-like substances and biosurfactants, which inhibit the growth and formation of biofilms [11,12].” – but growth of what, formation of biofilms - but who/what forms this biofilm.

The form of presentation of obtained results is an important advantage of the manuscript, the text of manuscript is prepared carefully,

Final decision – minor revision.

Author Response

Response to Reviewer 1:

Reviewer: I have read the manuscript with great interest. In my opinion the authors performed fully professional bioinformatics analysis of genomes of three strains of C. amycolatum strains isolated from vaginal biotope of healthywomen. I do not have any critical remarks about methodology – as I wrote above it is a really professional analysis. Some results are promising (from practical point of view – possibilities of application of isolated strain/strains as probiotics) and interesting (from scientific point of view). The strain that is (probably) able to produce bacteriocins seems to particularly interesting. Identification of some other genes is also very important – e.g. genes responsible for production of hydrogen peroxide and antibiotic resistance. Moreover, the authors revealed that the isolated bacteria do not produce “important” virulence factors, which is also very important.  However, I have also some critical remarks about the study. First of all I regret that the authors did not perform any experiments that could finally confirm antibiotic resistance or production of bacteriocins. I believe that the authors are going to perform these analysis in the near future. I also would be grateful for additional comment about possibilities of application of isolated strains – the strains were isolated from vaginal environment and I understand that they are able to survive under conditions that are typical for this biotope, but what about skin or gastrointestinal tract (are they able to survive under these conditions) – some probiotics are important for protection our skin and gastrointestinal tract against development  of pathogenic bacteria (e.g. during antibiotic therapies). In introduction the authors have written: “In addition, a large group of nondiphtheria corynebacteria is part of the resident microflora of human skin and mucous membranes and most often stands out from clinical samples”.

Authors: Dear Reviewer,

Thank you for your assessment of our manuscript and comments.

Recently, we have published an article containing the experimental results confirming  features of the strains described in this manuscript. The findings were published in the following paper:

Gladysheva, I.V., Chertkov, K.L., Cherkasov, S.V. et al. Probiotic Potential, Safety Properties, and Antifungal Activities of Corynebacterium amycolatum ICIS 9 and Corynebacterium amycolatum ICIS 53 Strains. Probiotics & Antimicro. Prot. (2021). https://doi.org/10.1007/s12602-021-09876-3

Our results showed that the studied strains suppress the growth and biofilm formation of pathogenic bacteria, as well as vaginal and intestinal isolates of Candida fungi. High levels of acid-tolerance and survival of the studied strains in the medium with bile salts suggest that the studied strains can successfully survive stomach and small intestine conditions, and colonize the gastrointestinal tract.

Below is an abstract of the article.

The purpose of this study was to evaluate the probiotic characteristics and safety and to study the antifungal activity of C. amycolatum ICIS 9 and C. amycolatum ICIS 53 against Candida spp. The probiotic potential and safety properties were assessed by standard parameters. Both strains showed good survival at pH 3 for 3 h and high tolerance to 0.3% bile salts after 4 h of incubation. The indicators of hydrophobicity, autoaggregation, and surface tension for ICIS 9 were 89.43% (n-hexane) and 73.96% (xylene) and ranged from 13.13% to 39.86% and 34.27 mN/m, respectively. For ICIS 53, they were 59.95% (n-hexane) and 45.68% (xylene), from 35.58% to 51.53% and 32.40 mN/m, respectively. The strains ICIS 9 and ICIS 53 exhibited varying levels of coaggregation with all eight examined bacterial pathogens. The ICIS 9 strain was resistant to amikacin, amoxicillin, clarithromycin, chloramphenicol, ciprofloxacin, and gentamycin. ICIS 53 was resistant only to ciprofloxacin. The cell-free supernatant of strains ICIS 9 and ICIS 53 showed good antimicrobial and antibiofilm activity against ten pathogenic vaginal and intestinal isolates of Candida spp. The CFS of ICIS 9 was more active against intestinal isolates, and the CFS of ICIS 53 showed good antimicrobial activity against vaginal isolates while inhibiting the growth of 2 out of 5 Candida spp. isolated from the intestine. Both of the strains were capable of reducing biofilm formation of Candida fungi. In the case of the vaginal isolates of C. krusei V1, the results showed that the inhibition levels of ICIS 9 and ICIS 53 were 36.75% and 11.4%, respectively. In the case of C. albicans (V2, V3, V7, V8), the inhibition of biofilm formation was no more than 7.07%. ICIS 9 and ICIS 53 also significantly inhibited biofilm formation of C. krusei 2613 intestinal isolates by 42.75% and 41.87%, respectively, with ICIS 9 inhibiting biofilm formation of C. albicans (2607, 2311, 2615, 2615) from 3.38% to 15.69% and ICIS 53 from 5.95% to 23.48%. None of the strains showed DNase, haemolytic or gelatinase activities. The results obtained revealed that ICIS 9 and ICIS 53 have safe properties and have the potential to be developed as probiotics.

We are currently planning experiments to determine the production of secondary metabolites and bacteriocins in the isolated strains.

Reviewer: Less important comments.

In some brackets the authors have written “Figura” instead of Figure.

Authors: corrected.

Reviewer: I The authors should use italic for presenting names of bacteria – please go through the whole text and write the names of microorganisms according to generally accepted rules.

Authors: corrected.

Reviewer: In introduction, the authors have written:” It was shown that certain types of nondiphtheria  corynebacteria produce various bacteriocins, bacteriocin-like substances and biosurfactants, which inhibit the growth and formation of biofilms [11,12].” – but growth of what, formation of biofilms - but who/what forms this biofilm.

Authors: corrected as following.

 “It was shown that certain types of nondiphtheria corynebacteria produce various bacteriocins, bacteriocin-like substances and biosurfactants, which inhibit the growth of opportunistic microorganisms and their biofilm formation [11,12].”

Reviewer: The form of presentation of obtained results is an important advantage of the manuscript, the text of manuscript is prepared carefully,

Final decision – minor revision.

Authors: Thank you so much!

Reviewer 2 Report

The authors reported the isolation of three strains of Corynebacterium amycolatum (ICIS5, ICIS9, and ICIS53) from the vaginal smear of healthy women, which may produce metabolites to keep healthy vaginal environments. For developing probiotics, secondary metabolites produced by bacteria have been investigated to prevent the growth of pathogenic microorganisms. Interestingly, non-diphtheriae Corynebacterium species are one of the major components in the vaginal biotope, and the authors suggested in this study that these species have several stress-resistance genes in their genomes to survive the aggressive vaginal environments, and bacteriocin-like, including sactipeptides, produced by Corynebacteria are possible substances for maintaining a healthy vaginal microbiome. The conclusion is reasonable, and I hope the following comments will improve this manuscript.

Major comments:

The most appealing finding in this study is secondary metabolites from Corynebacterium amycolatum for the healthy vaginal biotope. Since the authors’ group discovered sactipeptides from C. amycolatum previously, this point should be significantly emphasized. The paragraph (Lines 320-331) “Sactipeptides are a new class of ribosomally synthesized and …” will be nicer if it explains more about the sactipeptides from C. amycolatum. Several sactipeptides have been reported mainly from Bacillus, Ruminococcus, and Streptococcus. How is the sactipeptides from C. amycolatum ICIS9 different from the previously reported sactipeptides?

Corynebacteria are known to have unique transporters to secrete metabolites, such as amino acids. If only the strain ICIS9 has the ability to secrete Sactipeptides whereas the other strains do not, their transporters might be specialized for that aim. The list of transporter genes among the three strains would be interesting to compare.

Minor comments:

Introduction, lines 56-58: Ref 19 (Hammerschlag MR, Alpert S, Rosner I, et al. Microbiology of the vagina in children, 1978) reported that Corynebacterium vaginale is one of the major components in the isolates from vaginal cultures. Is C. amycolatum related to Corynebacterium vaginale? If so, the author should include Corynebacterium vaginale in the phylogenetic analysis.

Introduction, lines 71-72: The authors should provide the reason why they focused on Corynebacterium amycolatum mostly among other Corynebacterium species in the introduction section.

Material & methods, line76-77: Sampling information should be more detailed. It is important to provide the detailed condition of sampling for experimental reproductivity.

Result, table 1, line148-149: What does the difference in the N50 and L50 among the strains mean in table 1? It looks like the values of N50 and L50 of the ICIS9 strain are different from the others. Additionally, they are confusing because N50 and L50 in the table are 45,496 and 18, but they are described as 50,876 and 7 in the result section. The authors should double-check these values.

All figure legends need to be more detailed for readers for better understanding.

Author Response

Response to Reviewer 2:

Reviewer: The authors reported the isolation of three strains of Corynebacterium amycolatum (ICIS5, ICIS9, and ICIS53) from the vaginal smear of healthy women, which may produce metabolites to keep healthy vaginal environments. For developing probiotics, secondary metabolites produced by bacteria have been investigated to prevent the growth of pathogenic microorganisms. Interestingly, non-diphtheriae Corynebacterium species are one of the major components in the vaginal biotope, and the authors suggested in this study that these species have several stress-resistance genes in their genomes to survive the aggressive vaginal environments, and bacteriocin-like, including sactipeptides, produced by Corynebacteria are possible substances for maintaining a healthy vaginal microbiome. The conclusion is reasonable, and I hope the following comments will improve this manuscript.

Authors: Dear Reviewer,

Thank you for your assessment of our manuscript and comments.

Reviewer: Major comments:

The most appealing finding in this study is secondary metabolites from Corynebacterium amycolatum for the healthy vaginal biotope. Since the authors’ group discovered sactipeptides from C. amycolatum previously, this point should be significantly emphasized. The paragraph (Lines 320-331) “Sactipeptides are a new class of ribosomally synthesized and …” will be nicer if it explains more about the sactipeptides from C. amycolatum. Several sactipeptides have been reported mainly from Bacillus, Ruminococcus, and Streptococcus. How is the sactipeptides from C. amycolatum ICIS9 different from the previously reported sactipeptides?

Authors: Dear Reviewer,

Unfortunately, current study cannot give sufficient data for characteristics sactipeptide of C. amycolatum ICIS9 and its differences from the previously reported sactipeptides. Production of sactipeptides by corynebacteria has not yet been described in available literature. Moreover, we did not try to get sactipeptide from C. amycolatum ICIS9 and characterize it. In this study we managed to predict a gene encoding a sactipeptide in genome of C. amycolatum ICIS9  using the BAGEL 3 program. Based on the all body of current literature devoted to corynebacteria, we are sure that this report is the first fact indicating the production of sactipeptides by corynebacteria. But for description and comparable study we are going to evaluate further the production of this bacteriocin by the strain ICIS 9 in detail.

Reviewer: Corynebacteria are known to have unique transporters to secrete metabolites, such as amino acids. If only the strain ICIS9 has the ability to secrete Sactipeptides whereas the other strains do not, their transporters might be specialized for that aim. The list of transporter genes among the three strains would be interesting to compare.

Authors: Thanks a lot for the idea. Unfortunately, estimation of transporters was not in a focus of this study, so we did not analyse this issue. But we will take your recommendation into account in our further research.

Reviewer: Minor comments:

Introduction, lines 56-58: Ref 19 (Hammerschlag MR, Alpert S, Rosner I, et al. Microbiology of the vagina in children, 1978) reported that Corynebacterium vaginale is one of the major components in the isolates from vaginal cultures. Is C. amycolatum related to Corynebacterium vaginale? If so, the author should include Corynebacterium vaginale in the phylogenetic analysis.

Authors:

Dear Reviewer,

Based on the available literature, we can state that Corynebacterium vaginale was mentioned  till  1980. Since 1981, this species has not been occuring in research articles. Moreover, this species has been never described as valid in terms of ICNB; it is absent in microbiological collections and genetic databases, including GenBank (NCBI). To our minds, Corynebacterium vaginale  might be redescribed after 1980 as another species or even genus.

Reviewer: Introduction, lines 71-72: The authors should provide the reason why they focused on Corynebacterium amycolatum mostly among other Corynebacterium species in the introduction section.

Authors:  corrected as following:

Corynebacterium amycolatum was isolated for the first time by Collins and Burton from clinical specimens in 1988 [reference]. Based on our observations, C. amycolatum is rather frequently isolated from vaginal biotopes of healthy women, and features a high probiotic potential. Particularly, we isolated three strains of corynebacteriafrom the vaginal contents of healthy women. All of them were identified as C. amycolatum. Metabolites of these strains greatly increased the antagonistic activity of peroxide-producing lactobacilli against pathogenic and opportunistic microorganisms and had strong antimicrobial activity against opportunistic pathogens such as Escherichia coli, Staphylococcus aureus, Klebsiella pneumoniae and Pseudomonas aeruginosa [23,24]. Collins MD, Burton RA, Jones D (1988) Corynebacterium amycolatum sp. nov., a new mycolic acid-less Corynebacterium species from human skin. FEMS Microbiol Lett 49(3):349-352. https://doi.org/10.1111/j.1574-6968.1988.tb02755.x.

Reviewer: Material & methods, line76-77: Sampling information should be more detailed. It is important to provide the detailed condition of sampling for experimental reproductivity.

Authors: corrected as following:

Strains of C. amycolatum ICIS 5, ICIS 9 and ICIS 53, were previously isolated from vaginal smears of healthy women of reproductive age. The strains are deposited in the Collection of Microorganisms of the Institute for Cellular and Intracellular Symbiosis UrB RAS (Orenburg, Russia) under the same accession names. The phenotypic characteristics of these isolates have been previously described in detail [23,24]. The strains were kept at − 80 °C in 20% (v/v) glycerol before experiment. The isolates were grown in tryptic soy broth (TSB) at 37 °C for 24 h.

Because for high-throughput sequencing we took the strains from local microbial collection, we did not give detailed description of sampling indicated below. But if the Reviewer finds this essential for the manuscript, we will include the procedure description in the text.

Earlier, samples were collected by setting up a sterile speculum without antiseptic cleaning of the exocervix, and a swab was inserted into the endocervix by performing a rotational movement. Samples from each patient were taken using a Sigma Transwab (Medical Wire, Corsham, United Kingdom).  Then, the swab was introduced into sterile tryptone–salt (TS) solution (Sigma-Aldrich, Steinheim, Germany) prior further analyses.  One milliliter of the collected swab was introduced into 3 ml of tryptic soy broth (TSB) (Becton Dickinson, USA)  and incubated for 24 h at 37 °C. After this period, Tryptic Soy Agar (Becton Dickinson, USA)  plates were inoculated and incubated at 37 °C for 24–48 h. After incubation, the bacterial colonies were inspected and isolated.

Reviewer: Result, table 1, line148-149: What does the difference in the N50 and L50 among the strains mean in table 1? It looks like the values of N50 and L50 of the ICIS9 strain are different from the others..

Authors: N50 and L50 are characteristics of genome assembly. The assembly of ICIS9 genome in terms of N50 and L50 is more fragmented than ICIS 5 and ICIS 53. But in terms of gene number all genome assemblies are similar that allows their comparing.

Reviewer: Additionally, they are confusing because N50 and L50 in the table are 45,496 and 18, but they are described as 50,876 and 7 in the result section. The authors should double-check these values.

Authors: Thanks for this finding!  We have checked the values. Indeed, we made a misprint. It was corrected.

Reviewer: All figure legends need to be more detailed for readers for better understanding.

Authors: corrected as following:

Figure 1. Phylogenetic tree highlighting the position of C. amycolatum ICIS 5, ICIS 9 and ICIS 53 relative to the 28 type strains of Corynebacterium spp. most often isolated from human clinical samples and 2 strains of Euzebya spp. as an outgroup. The phylogenetic tree was constructed by MrBayes V. 3.27 using the GTR replacement model on the Unipro UGENE software platform (version 35.0). Corresponding NCBI accession numbers are shown in parentheses.